# Modeling Noisy Annotations for Crowd Counting

**Jia Wan**    **Antoni B. Chan**
Department of Computer Science
City University of Hong Kong
jiawan1998@gmail.com, abchan@cityu.edu.hk

## Abstract

The annotation noise in crowd counting is not modeled in traditional crowd counting algorithms based on crowd density maps. In this paper, we first model the annotation noise using a random variable with Gaussian distribution, and derive the pdf of the crowd density value for each spatial location in the image. We then approximate the joint distribution of the density values (i.e., the distribution of density maps) with a full covariance multivariate Gaussian density, and derive a low-rank approximate for tractable implementation. We use our loss function to train a crowd density map estimator and achieve state-of-the-art performance on three large-scale crowd counting datasets, which confirms its effectiveness. Examination of the predictions of the trained model shows that it can correctly predict the locations of people in spite of the noisy training data, which demonstrates the robustness of our loss function to annotation noise.

## 1 Introduction

Crowd counting is an important research topic that can be used in monitoring the crowd behaviors for public safety. Most crowd counting algorithms utilize a deep network to predict density maps from crowd images, where the sum over a region in the density map is the prediction of the count in that region. Most methods treat this problem as a standard regression task, using the standard L2 norm between the ground-truth and predicted density maps as the loss function [1, 2, 3]. Besides L2-norm, Bayesian loss (BL) [4] consider an annotation-wise loss function, based on point supervision.

However, both L2 and BL have two deficiencies. First, the noise in the annotation process is not considered in a principled way. L2 and BL make an assumption about per-pixel i.i.d. Gaussian additive noise on the observed density value. However, the noise in the density map arises from the noise in the human annotations (i.e., the displacement of the annotated locations), which in general should not lead to i.i.d Gaussian noise observation noise. Second, the correlation between pixels is ignored. L2 and BL assume independence between pixels in the density map. However, if an annotation moves, the changes in the density map in nearby pixels are correlated.

To address these issues, we proposed a novel loss function that explicitly models annotation noise as random variables. Instead of considering the human annotations as the "ground-truth" locations, we consider the annotations as noisy observations of the true person locations. The annotation noise around the true locations induces a joint probability distribution over density maps. First, we derive the form of the marginal probability distribution of the crowd density value at a spatial location in the density map. However, due to its intractability, we propose to approximate this marginal distribution using a Gaussian distribution, leading to an approximation to the joint probability density based on products of marginals. In order to model correlations between pixels, we then derive an approximation to the *joint* probability distribution using a multivariate Gaussian distribution with full covariance matrix. As the multivariate Gaussian has dimension equal to the size of the image, we then propose a low-rank method to approximate the covariance matrix, which reduces computational

complexity. Finally, the negative log-likelihood of the m.v. Gaussian is used as the loss function for training the density map estimator.

The proposed loss function decomposes into a per-pixel weighted L2-norm term and a correlation term. In the per-pixel weighted term, the loss will reduce the weight for uncertain regions, as compared to the L2 loss (which uses equal weights). Therefore, the proposed loss function is robust to annotation noise and the predicted high-density regions are allowed to move around to find a consistent location. The second term models the correlations between pixels in high-density regions, which can improve training of the density map estimator.

The contributions of this paper are three-fold: 1) We propose to explicitly model annotation noise as a random variable and derive the corresponding probability density function of crowd density maps; 2) We derive a m.v. Gaussian approximation to this probability density, whose negative log-likelihood is used as a loss function for training deep networks; 3) We derive a low-rank covariance approximation to improve the efficiency of calculating the negative log-likelihood for practical application.

## 2    Related works

Traditional crowd counting algorithms can be divided into two categories: counting by detection and counting by global regression. [5] proposes a head and shoulder detector for crowd counting which will fail for high density and occluded images. To avoid explicit detection of individuals, global regression algorithms are proposed [6] to direct predict global count based on low-level features [7]. Multiple features are fused in [8] to further improve the performance. However, those methods are hard to deal with large scale variation and occlusion in high density crowds.

### 2.1    Density map based crowd counting

To handle challenges in high density crowds, density map based algorithms are proposed and achieve remarkable improvement in recent years. Those algorithms first generate the ground-truth density maps from annotation dot maps [9], typically by convolving with a Gaussian kernel with fixed [9] or adaptive bandwidth [10, 11]. Then, different network structures are proposed to deal with challenges in crowd counting, such as scale variation and perspective distortion.

To extract multi-scale features, [11] proposes a Multi-column Neural Network (MCNN) in which different columns have different receptive fields. Switch-CNN [12] proposes to select column with proper receptive field instead of using features from all scales. [13] proposes a tree-structured CNN to solve scale variation. [14] proposes a consistency constraint to deal with inconsistency across different scales. [3] proposes to extract multi-scale features for all convolution layers. [15] propose to use image pyramid and a density based attention to merge features from different scales. [16] propose a pass message across different scales using a CRF based structure.

The quality of density maps is improved using refinement and ensemble based methods. [17] propose to generate high-resolution density map based on the low-resolution density map generated in the initialized stage. [18] proposes a top-down feedback strategy to refine the predicted density map. A region-based refinement approach is proposed to in [19]. Ensemble based approaches are also effective to improve the density map quality. [20] uses a boosting mechanism and [21] uses multiple negative correlated regressors to obtain high-quality density maps.

Finally, crowd counting performance can be improved by exploiting context information. [22] proposes a Contextual Pyramid CNN (CP-CNN), while [23] uses temporal context. [24] propose to utilize unlabeled images using a ranking-based method. A composition loss [25] is proposed to solve counting, density map estimation and localization simultaneously. Other works shown that density map is also useful for pedestrian detection and tracking in crowd images [26, 27, 28].

Most of the density map based methods use L2 norm as the loss function, which assumes i.i.d. Gaussian noise for the observed density value. In contrast, our proposed loss can model the fundamental annotation noise according to a generative process, as well as the correlations between pixels.

### 2.2    Dot-map based crowd counting

Density maps are essentially intermediate representations that are constructed from an annotation dot map, indicating the observed locations of each person in the image. Most methods use a fixed

Gaussian [9] or adaptive Gaussian kernels [10, 11], but the optimal choice of bandwidth/method varies with the dataset and network architecture [2]. To address this issue, [2] proposes a framework to jointly generate and estimate density maps in an end-to-end manner. [29] proposes to count via localization, while [30] proposes a detection-like framework based on dot annotations. Such frameworks need additional networks such as detectors or localization networks. Bayesian loss (BL) [4] is an annotation-wise loss function based on point supervision. Although BL does not require a density map, it ignores the annotation noise and correlations, similar to the standard L2 norm. Thus, we propose a novel loss function that more naturally represents the annotation noise and correlations among pixels. Our loss function can be applied to all density map based network architectures, and we evaluate it using three different backbones, VGG19 [4], CSRNet [1] and MCNN [11], and achieve superior performance compared to other loss functions.

## 2.3 Uncertainty modeling

Uncertainty modeling is adopted in computer vision tasks to improve the robustness of algorithms to label noise [31]. [32] uses dropout to model the uncertainty of the prediction for semantic segmentation. [33] proposes to model the face representation and the uncertainty simultaneously. However, most of these works are proposed to model label or feature uncertainty, while the spatial annotation noise is not considered, although it is common in tasks like crowd counting. To improve the robustness of counting algorithms, we propose to model spatial-noise in the annotated people locations explicitly.

## 3 Methodology

In this section, we first review the traditional density map method, followed by our proposed model using noisy annotations.

## 3.1 Traditional density map method

We first review the traditional method for generating density maps [9]. For a given image $\mathcal{I}$, let there be $N$ annotations $\{\tilde{\mathbf{D}}_i\}_{i=1}^{N}$, corresponding to the observed location of each person in the image. For a given spatial location $\mathbf{x}$ in the image, the corresponding density value $y$ at pixel location $\mathbf{x}$ is obtained by placing a Gaussian kernel at each annotation,

$$y(\mathbf{x}) = \sum_{i=1}^{N} \mathcal{N}(\mathbf{x}|\tilde{\mathbf{D}}_i, \beta\mathbf{I}) = \sum_{i} \frac{1}{2\pi\beta} \exp(-\frac{1}{2}||\mathbf{x} - \tilde{\mathbf{D}}_i||^2_{\beta\mathbf{I}}), \tag{1}$$

where $\beta$ is the variance of the Gaussian kernel, and $\mathcal{N}(\mathbf{x}|\boldsymbol{\mu}, \boldsymbol{\Sigma})$ is the pdf for a multivariate Gaussian with mean $\boldsymbol{\mu}$ and covariance matrix $\boldsymbol{\Sigma}$, with $||\mathbf{x}||^2_{\boldsymbol{\Sigma}} = \mathbf{x}^T\boldsymbol{\Sigma}^{-1}\mathbf{x}$ as the squared Mahalanobis distance. Computing (1) for all locations in the image yields a density map $\mathbf{y}$ for the image $\mathcal{I}$ (see Figure 1), where the sum inside a region is the crowd count.

The goal of crowd counting is then to predict the density map $\mathbf{y}$ from the image $\mathcal{I}$, which is framed as learning the regressor $f(\mathcal{I})$ using the L2-norm loss, $\mathcal{L}(\mathbf{y}, f(\mathcal{I})) = ||\mathbf{y} - f(\mathcal{I})||^2$. Note that a standard result in machine learning [34] is that learning with L2 loss is equivalent to assuming i.i.d Gaussian noise between the underlying function (in this case $f(\mathcal{I})$) and its observations (in this case $\mathbf{y}$). However, such an assumption is tenuous, as the observation noise is induced by the noise in the observed annotation locations $\tilde{\mathbf{D}}$ that is passed through the non-linear function in (1).

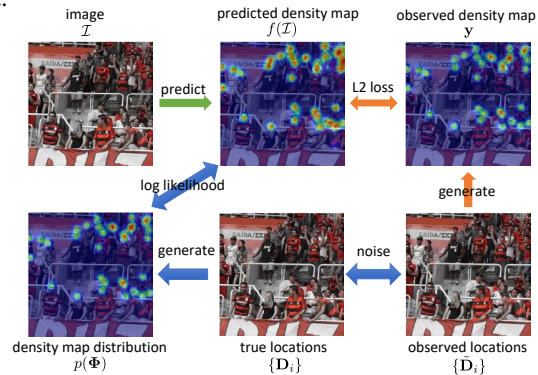

Figure 1: Our framework for modeling noisy annotations. Blue arrows are our model, and orange arrows are the traditional density map generation method.

## 3.2 Modeling noisy annotations

We now consider that the annotations $\{\tilde{\mathbf{D}}_i\}$ are actually noisy observations about the locations of the people in the image (see Fig. 1). Let $\mathbf{D}_i$ be the true location of the i-th person, and thus

$\mathbf{D}_i = \tilde{\mathbf{D}}_i + \boldsymbol{\epsilon}_i$, where $\boldsymbol{\epsilon}_i$ is the annotation noise. We assume that the annotation noise is i.i.d. m.v. Gaussian, $\boldsymbol{\epsilon}_i \sim \mathcal{N}(0, \alpha\mathbf{I})$, where $\alpha$ is the variance parameter. Using the true annotations $\mathbf{D}_i$, the density value at location $\mathbf{x}$ is

$$\Phi = \sum_{i=1}^{N} \mathcal{N}(\mathbf{x}|\mathbf{D}_i, \beta\mathbf{I}) = \sum_{i=1}^{N} \mathcal{N}(\mathbf{x}|\tilde{\mathbf{D}}_i + \boldsymbol{\epsilon}_i, \beta\mathbf{I}) = \sum_{i=1}^{N} \mathcal{N}(\mathbf{q}_i|\epsilon_i, \beta\mathbf{I}) \triangleq \sum_{i} \phi_i, \qquad (2)$$

where $\phi_i$ is the individual term for each annotation, and $\mathbf{q}_i = \mathbf{x} - \tilde{\mathbf{D}}_i$ is the difference between the location of the i-th annotation and location $\mathbf{x}$. Note that, since $\{\boldsymbol{\epsilon}_i\}$ are random variables, then density value $\Phi$ is also a random variable. Finally, collecting the density value r.v. for each location $\mathbf{x}$, we obtain a multivariate r.v. for the density map of the image, $\boldsymbol{\Psi} = [\Phi^{(1)}, \cdots, \Phi^{(P)}]$, where the entries are (2) evaluated at all locations $\mathbf{x}^{(1)}, \cdots, \mathbf{x}^{(P)}$ of the image. As multivariate r.v. $\boldsymbol{\Psi}$ is complex, we first derive a form of its marginal distribution $\Phi$ at location $\mathbf{x}$ and an efficient Gaussian approximation. We then consider a m.v. Gaussian approximation to the joint likelihood of $\boldsymbol{\Psi}$.

### 3.2.1 Probability distribution of $\Phi$

We consider the marginal of $\boldsymbol{\Psi}$, which corresponds to the pdf of $\Phi$ at a specific location $\mathbf{x}$. The form of the pdf of $\Phi$ can be derived by passing the Gaussian r.v.s $\{\boldsymbol{\epsilon}_i\}$ through the non-linear transformation in (2). First looking at the individual term $\phi_i = \frac{1}{2\pi\beta}\exp(-\frac{1}{2}\|\mathbf{q}_i - \epsilon_i\|_{\beta\mathbf{I}}^2)$, it consists of a series of transformations of a multivariate Gaussian r.v.: squared L2 norm, negative exponential, and scaling. Noting that the squared L2 norm of a m.v. Gaussian random variable with non-zero mean is a non-central $\chi^2$ distribution, and applying formulas for the transformation of a random variable (see Supp. for derivation) we obtain the density of $\phi_i$ as

$$\phi_i \sim p_i(\phi_i) = \frac{\delta}{h}e^{-\lambda_i/2}\left(\frac{\phi_i}{h}\right)^{\delta-1} I_0(\sqrt{-2\delta\lambda_i \log \frac{\phi_i}{h}}), \qquad (3)$$

where $h = (2\pi\beta)^{-1}$ is the maximum value of the Gaussian kernel, $\delta = \beta/\alpha$, and $I_0(x)$ is a modified Bessel function of the 1st kind of order 0. Second, since the noise $\boldsymbol{\epsilon}_i$ for each annotation are assumed independent, the resulting individual terms $\phi_i$ are also independent r.v.s. Thus the pdf of the sum $\Phi = \sum_i \phi_i$ is the convolution of the pdfs of the individual terms, $\Phi \sim p(\Phi) = p_1(\Phi)*p_2(\Phi)*\cdots*p_N(\Phi)$. However, this convolution is intractable to compute in closed form.

### 3.2.2 Gaussian approximation to $\Phi$

We approximate the distribution of $\Phi$ using a Gaussian, $\hat{p}(\Phi) = \mathcal{N}(\Phi|\mu, \sigma^2)$, where $\mu$ and $\sigma^2$ are mean and variance of the distribution. The mean of $\Phi$ is calculated as (see derivation in Supp.)

$$\mu = \mathbb{E}[\Phi] = \mathbb{E}[\sum_i \mathcal{N}(\mathbf{q}_i|\boldsymbol{\epsilon}_i, \beta\mathbf{I})] = \sum_i \mathcal{N}(\mathbf{q}_i|\mathbf{0}, (\alpha+\beta)\mathbf{I}) \triangleq \sum_i \mu_i, \qquad (4)$$

where $\mu_i$ is the mean for the individual term $\phi_i$, and the variance is

$$\sigma^2 = \text{var}(\Phi) = \mathbb{E}[\Phi^2] - \mathbb{E}[\Phi]^2 = \sum_i \frac{1}{4\pi\beta}\mathcal{N}(\mathbf{q}_i|\mathbf{0}, (\beta/2+\alpha)\mathbf{I}) - \sum_i \mu_i^2. \qquad (5)$$

Figure 2 (a-c) shows an example with three annotations and the corresponding marginal distributions of $\Phi$ for two spatial locations.

We use a Gaussian approximation since it is tractable and can be estimated from the 1st and 2nd moments of $\Phi$. Extensions of the central limit theorem prove that sums of independent non-identical r.v.s converge to Gaussian. Indeed, in Fig. 2 (c), the distribution is tending to Gaussian with just 3 annotations, and we observe this tendency becomes stronger with more annotations. We have also tried Gamma distributions for the approximation, but the results are worse (MAE is 89.7 on UCF-QNRF).

### 3.2.3 Gaussian approximation to joint likelihood of $\boldsymbol{\Psi}$

The previous derivation approximates each spatial location $\mathbf{x}$ independently. We next consider the case that models correlation between locations via a m.v. Gaussian approximation to the joint likelihood $\boldsymbol{\Psi}$. Given the spatial locations $\mathbf{x}^{(\eta)}$, for $\eta \in \{1, \cdots, P\}$, let $\mathbf{q}_i^{(\eta)} = \mathbf{x}^{(\eta)} - \tilde{\mathbf{D}}_i$ be the

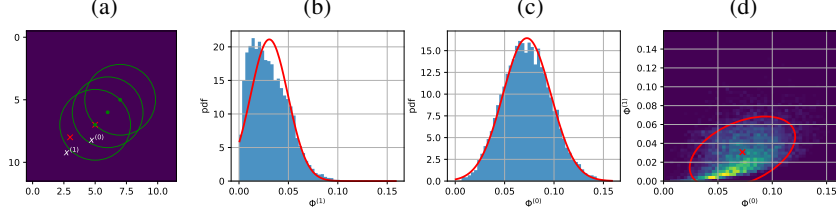

Figure 2: Example of probability distributions of density values: (a) three annotations (green points), where the circles represent 2 standard deviations of the annotation noise; (b) marginal distribution of density values at location $\mathbf{x}^{(1)}$ and (c) $\mathbf{x}^{(0)}$; (d) the joint distribution of density values $(\Phi^{(1)}, \Phi^{(0)})$. The histograms in (b-d) are obtained through sampling, and the red lines are the Gaussian approximations.

difference between the spatial location of the i-th annotation and the pixel location $\mathbf{x}^{(\eta)}$. From (2), the density value r.v. $\Phi^{(\eta)}$ at pixel location $\mathbf{x}^{(\eta)}$ is

$$\Phi^{(\eta)} = \sum_{i=1}^{N} \mathcal{N}(\mathbf{q}_i^{(\eta)} | \boldsymbol{\epsilon}_i, \beta \mathbf{I}) \triangleq \sum_i \phi_i^{(\eta)}. \tag{6}$$

Note that the annotation noise $\boldsymbol{\epsilon}_i$ is the same r.v. across all $\Phi^{(\eta)}$.

We define the Gaussian approximation to $\boldsymbol{\Psi}$ as $\hat{p}(\boldsymbol{\Psi}) = \mathcal{N}(\boldsymbol{\Psi} | \boldsymbol{\mu}, \boldsymbol{\Sigma})$, where $(\boldsymbol{\mu}, \boldsymbol{\Sigma})$ are the mean vector and covariance matrix. From the previous derivation, we obtain the entries in $\boldsymbol{\mu}$ as $\mathbb{E}[\Phi^{(\eta)}] = \sum_i \mu_i^{(\eta)}$, which is computed with (4), and the diagonal of the covariance matrix as the variances $\boldsymbol{\Sigma}_{\eta,\eta} = \mathrm{var}(\Phi^{(\eta)})$ as in (5). The covariance terms are derived as (see Supp. for derivation),

$$\boldsymbol{\Sigma}_{\eta,\rho} = \mathrm{cov}(\Phi^{(\eta)}, \Phi^{(\rho)}) = \sum_i \omega_i^{(\eta,\rho)} - \sum_i \mu_i^{(\eta)} \mu_i^{(\rho)}, \tag{7}$$

where $\omega_i^{(\eta,\rho)} = \mathbb{E}[\phi_i^{(\eta)} \phi_i^{(\rho)}] = \mathcal{N}(\mathbf{x}^{(\eta)} | \mathbf{x}^{(\rho)}, 2\beta \mathbf{I}) \mathcal{N}(\frac{1}{2}(\mathbf{q}_i^{(\eta)} + \mathbf{q}_i^{(\rho)}) | \mathbf{0}, (\beta/2 + \alpha) \mathbf{I})$. Fig. 2 (d) shows an example of the joint distribution for two spatial locations, while Fig. 3 (top) shows an example on a small image.

### 3.2.4 Low-rank approximation to covariance matrix

The covariance matrix $\boldsymbol{\Sigma}$ has dimension $P \times P$, which does not scale well in computation/storage for large images. However, in $\boldsymbol{\Sigma}$, most of the off-diagonal elements in a row or column are zero if that spatial location is far from an annotation. Thus $\boldsymbol{\Sigma}$ can be approximated by only storing those rows/columns with significant covariance values (see full derivation in Supp.).

Let $\mathcal{M} = \{m_1, \cdots, m_M\}$ be the set of indices of spatial locations $x^{(m_i)}$ that we want to use for the approximation. The approximation to $\boldsymbol{\Sigma}$ only uses the off-diagonal elements corresponding to $\mathcal{M}$,

$$\boldsymbol{\Sigma} \approx \hat{\boldsymbol{\Sigma}} = \mathbf{V} + \mathbf{M} \mathbf{A}_{\mathcal{M}} \mathbf{M}^T, \tag{8}$$

where $\mathbf{V} = \mathrm{diag}(\mathrm{diag}(\boldsymbol{\Sigma}))$ is the diagonal matrix of the diagonal of $\boldsymbol{\Sigma}$, $\mathbf{M}$ is a permutation matrix with i-th column $[\mathbf{M}]_i = \mathbf{e}_{m_i}$, and the selected off-diagonal entries are

$$[\mathbf{A}_{\mathcal{M}}]_{ij} = \begin{cases} 0, & i = j, \\ \mathrm{cov}(\Phi^{(m_i)}, \Phi^{(m_j)}), & i \neq j. \end{cases} \tag{9}$$

Using the matrix inversion lemma, we obtain the approximate inverse covariance matrix, $\hat{\boldsymbol{\Sigma}}^{-1} = \mathbf{V}^{-1} - \mathbf{M} \mathbf{B}_{\mathcal{M}} \mathbf{M}^T$, where $\mathbf{B}_{\mathcal{M}} = (\mathbf{V}_{\mathcal{M}} \mathbf{A}_{\mathcal{M}}^{-1} \mathbf{V}_{\mathcal{M}} + \mathbf{V}_{\mathcal{M}})^{-1}$ and $\mathbf{V}_{\mathcal{M}} = \mathbf{M}^T \mathbf{V} \mathbf{M}$. Finally, the approximate negative log-likelihood function is (ignoring constant terms),

$$-\log \hat{p}(\boldsymbol{\Psi}) = -\log \mathcal{N}(\boldsymbol{\Psi} | \boldsymbol{\mu}, \hat{\boldsymbol{\Sigma}}) \propto ||\boldsymbol{\Psi} - \boldsymbol{\mu}||_{\hat{\boldsymbol{\Sigma}}}^2 = \bar{\boldsymbol{\Psi}}^T \mathbf{V}^{-1} \bar{\boldsymbol{\Psi}} - \bar{\boldsymbol{\Psi}}^T \mathbf{M} \mathbf{B}_{\mathcal{M}} \mathbf{M}^T \bar{\boldsymbol{\Psi}}, \tag{10}$$

where $\bar{\boldsymbol{\Psi}} = \boldsymbol{\Psi} - \boldsymbol{\mu}$. Since $\mathbf{V}$ is diagonal, the first term in (10) is equivalent to the sum over the negative log-marginals of $\Phi$ (i.e., a diagonal covariance matrix). The second term is the correlation term, based on the $M$ selected entries $\mathbf{M}^T \bar{\boldsymbol{\Psi}}$. The storage/computational complexity for one training example using the low-rank approximation is $O(M^2 + N)$ compared to $O(N^2)$ for the full covariance.

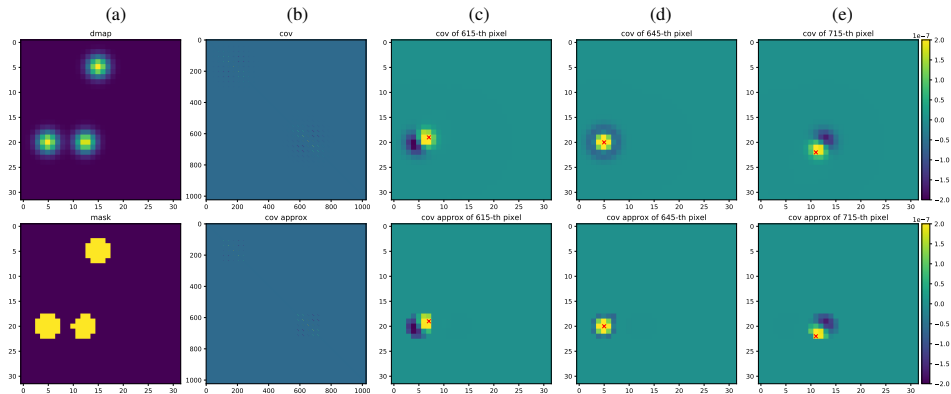

Figure 3: (top) Example Gaussian approximation to $\boldsymbol{\Psi}$: (a) mean vector (reshaped to $32\times32$); (b) covariance matrix; (c-e) covariance map for the marked spatial location (red x), corresponding to one row in the covariance matrix in (b) reshaped to $32\times32$. (bottom) low-rank approximation using the indices in the mask (bottom-a).

Table 1: Comparison of different loss functions with different backbones on UCF-QNRF.

|  | VGG19 [4] | | CSRNet [1] | | MCNN [11] | |
|---|---|---|---|---|---|---|
|  | MAE | MSE | MAE | MSE | MAE | MSE |
| L2 | 98.7 | 176.1 | 110.6 | 190.1 | 186.4 | 283.6 |
| BL [4] | 88.8 | 154.8 | 107.5 | 184.3 | 190.6 | 272.3 |
| Ours | **85.8** | **150.6** | **96.5** | **163.3** | **177.4** | **259.0** |

Table 2: Effect of each component on UCF-QNRF.

|  | MAE | MSE |
|---|---|---|
| L2 | 98.7 | 176.1 |
| L2+Reg | 94.5 | 160.0 |
| DiagCov | 103.2 | 175.8 |
| FullCov | 92.6 | 159.7 |
| FullCov + Reg | 89.9 | 155.9 |
| FullCov + Reg + bkg (Ours) | **85.8** | **150.6** |

Finally, for the selection of indices $\mathcal{M}$, note that the covariance is bounded by the corresponding variances, $\text{cov}(x,y) \leq \sqrt{\text{var}(x)\text{var}(y)}$, which implies that the largest covariance values will tend to be those associated with the largest variances. Thus, one selection criteria is to order the variances, $\mathcal{M} = \text{argmax}_\eta^M \boldsymbol{\Sigma}_{\eta,\eta}$, where the $\text{argmax}^M$ selects the top-$M$ indices. Here $M$ could be selected as a fixed value, or set adaptively to select a fixed percentage of variance, e.g., $\frac{\sum_{i=1}^M v_i}{\sum_{i=1}^P v_i} \approx 0.8$. Figure 3 shows an example of the low-rank approximation.

### 3.3 Regularization and background modeling

We use a regularization term, inspired by [4], to ensure that the predicted density map near each annotation satisfies the rule $\int p(x)\mathrm{d}x = 1$. For the i-th annotation point, we define the regularizer $\mathcal{L}_i^r = |\sum_\eta \Psi^{(\eta)} \frac{p(\phi_i^{(\eta)})}{\sum_{i=1}^N p(\phi_i^{(\eta)})} - 1|$, where $\Psi^{(\eta)}$ is the $\eta$-th entry of the predicted density map $\boldsymbol{\Psi}$. Then, the final loss function is: $\mathcal{L} = \lambda \bar{\boldsymbol{\Psi}}^T \hat{\boldsymbol{\Sigma}}^{-1} \bar{\boldsymbol{\Psi}} + \sum_{i=1}^N \mathcal{L}_i^r$.

In practice, the variance of a pixels far from the annotation point is close to 0, which will cause infinite loss. Thus, we propose to add a "virtual dot" to each spatial pixel which will prevent zero variance. The distance between the "virtual dot" and the $i$-th spatial pixel is defined as $d_i = d \cdot \exp \frac{-d_{min}}{\alpha+\beta}$, where $d$ is set to 15% of the shorter side of the image, as in [4]. $d_{min}$ is the distance between the pixel and its nearest annotation point. $d_i \approx 0$ for pixels far from the annotation point, while $d_i \approx d$ for pixels close to the annotation point. Therefore, the variance of pixels far from annotation points will be increased and the variance of pixels close to annotation points will not be impacted.

## 4 Experiments

In this section, we present experiments using our loss in (10) for training density map estimators.

### 4.1 Experimental setup

**Datasets:** The experiments are conducted on 6 datasets: NWPU-Crowd [35], JHU-CROWD++ [36], UCF-QNRF [25], Shanghai_Tech [11], UCSD [6], and Mall [37]. NWPU-CROWD is a large-scale benchmark for crowd counting which consists of 3,109 training images, 500 validation images and 1,500 testing images. JHU-CROWD++ has 4,371 images (2,722, 500, and 1,600 for train, val, test). UCF-QNRF contains 1,535 high-resolution images (1,201/334 for training/validation). Shanghai_Tech dataset consists of Part A and Part B. Part A has 482 and 300 images for training and

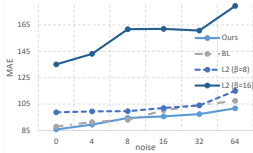

Figure 4: Robustness to annotation noise.

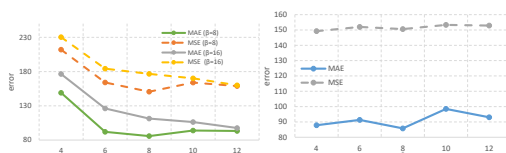

Figure 5: Effect of $\alpha$ and $\beta$.

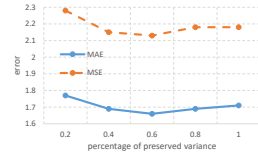

Figure 6: The effect of different preserved variances.

evaluation, while Part B has 716 and 400 images for training and testing. For the datasets without a validation set, we use 10% of the images for validation.

**Metrics:** Follow by previous works [38, 39], we use Mean Absolute Error (MAE) and Root Mean Squared Error (MSE) as the metric:

$$MAE = \frac{1}{N}\sum_i |\hat{y}_i - y_i|, \quad MSE = (\frac{1}{N}\sum_i \|\hat{y}_i - y_i\|^2)^{1/2},$$

where $N$ is the number of samples and $\hat{y}_i$, $y_i$ are the predicted and ground truth counts.

**Backbones and training:** We test 3 backbones: VGG19 [4], CSRNet [1], and MCNN [11]. VGG19 and CSRNet are pre-trained on ImageNet dataset, while MCNN is trained from scratch. We use Adam optimizer for training with learning rate $10^{-5}$. The regularization weight $\lambda$ is set to 0.1.

## 4.2 Ablation studies

We first conduct a series of ablation studies on our loss function.

### 4.2.1 Comparison of different loss functions

To evaluate the effectiveness of the proposed loss function, we compare it with L2, which is the most popular loss function, and the BL [4], which has the recent state-of-the-art performance. The results are shown in Tab. 1. The error of density estimators trained with our loss function is generally lower than those of other losses, for a variety of backbones. This demonstrates that modeling noisy annotations and correlation among pixels are important factors.

### 4.2.2 Robustness to annotation noise

Since the proposed loss function explicitly models the annotation noise, we conduct an experiment to verify its robustness to annotation noise. First, we generate a noisy dataset by randomly moving the annotation points by $\{4, 8, 16, 32, 64\}$ pixels. Note that the average head size is around 33 pixels. Second, we train the backbone using different loss functions on the noisy datasets. As shown in Fig. 4, the performance of three loss function all decrease as

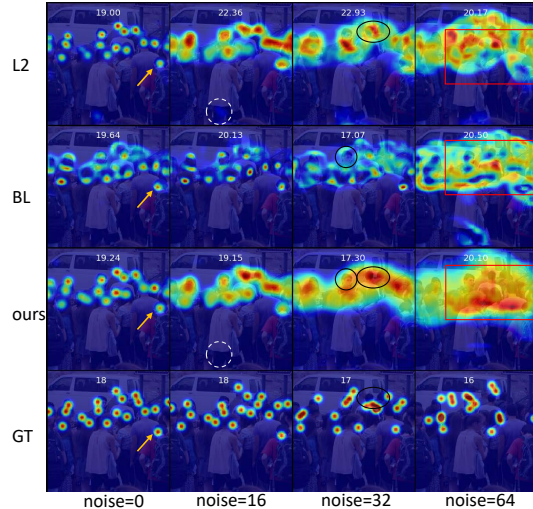

Figure 7: Visualization of density maps predicted from models trained with different loss functions and noise levels. More details can be found in the Supplemental.

the annotation noise increases. However, our proposed loss is more robust compared to other losses, showing a much lower MAE for large noise levels.

Fig. 7 shows the density maps predicted by models trained with our loss function. Although some noisy annotation points are not at the center of the head, our models can move the predicted density towards the head center, i.e., correct the annotation errors. When the annotations are very noisy, our predicted density maps are still smooth, showing that the model still learns under high uncertainty. See the supplemental for more detailed comparisons with other losses.

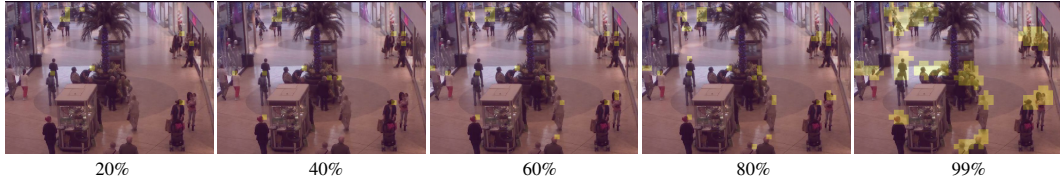

| 20% | 40% | 60% | 80% | 99% |

Figure 8: Example the selected locations $\mathcal{M}$ (highlighted yellow) for different percentage preserved variance preserved in the Mall dataset.

Table 3: Comparison with state-of-the-art crowd counting methods.

|  | NWPU-Crowd | | JHU-CROWD++ | | UCF-QNRF | | Shanghai_Tech A | | Shanghai_Tech B | | UCSD | |
|---|---|---|---|---|---|---|---|---|---|---|---|---|
|  | MAE | MSE | MAE | MSE | MAE | MSE | MAE | MSE | MAE | MSE | MAE | MSE |
| Uncertainty [31] | - | - | - | - | 103.2 | 103.2 | - | - | - | - | - | - |
| CP-CNN [22] | - | - | - | - | - | - | 73.6 | 106.4 | 20.1 | 30.1 | - | - |
| ASACP [14] | - | - | - | - | - | - | 75.7 | 102.7 | 17.2 | 27.4 | 1.04 | 1.35 |
| Switch-CNN [12] | - | - | - | - | 228.0 | 445.0 | 90.4 | 135.0 | 21.6 | 33.4 | 1.62 | 2.10 |
| CMTL [41] | - | - | 157.8 | 490.4 | 252.0 | 514.0 | 101.3 | 152.4 | 20.0 | 31.1 | - | - |
| CL [25] | - | - | - | - | 132.0 | 191.0 | - | - | - | - | - | - |
| LSCCNN [42] | - | - | 112.7 | 454.4 | 120.5 | 218.2 | 66.5 | 101.8 | 7.7 | 12.7 | - | - |
| MCNN [11] | 232.5 | 714.6 | 188.9 | 483.4 | 277.0 | 426.0 | 110.2 | 173.2 | 26.4 | 41.3 | 1.07 | 1.35 |
| CSRNet [1] | 121.3 | **387.8** | 85.9 | 309.2 | 110.6 | 190.1 | 68.2 | 115.0 | 10.6 | 16.0 | 1.16 | 1.47 |
| SANet [3] | 190.6 | 491.4 | 91.1 | 320.4 | - | - | 67.0 | 104.5 | 8.4 | 13.6 | 1.02 | **1.29** |
| DSSINet [16] | - | - | 133.5 | 416.5 | 99.1 | 159.2 | 60.6 | 96.0 | **6.8** | **10.3** | - | - |
| MBTTBF [43] | - | - | 81.8 | 299.1 | 97.5 | 165.2 | **60.2** | **94.1** | 8.0 | 15.5 | - | - |
| BL [4] | 105.4 | 454.2 | 75.0 | 299.9 | 88.7 | 154.8 | 62.8 | 101.8 | 7.7 | 12.7 | - | - |
| Ours | **96.9** | 534.2 | **67.7** | **258.5** | **85.8** | **150.6** | 61.9 | 99.6 | 7.4 | 11.3 | **1.00** | 1.35 |

### 4.2.3 Effect of variances $\alpha, \beta$

We conduct an experiment to investigate the effect of annotation variance $\alpha$ and the density map variance $\beta$. The results are shown in Fig. 5. First, the MAE is large when $\alpha$ is very small, and the MAE decreases and stabilizes when $\alpha \geq 8$. Therefore, we can assume that the annotation noise of UCF-QNRF is 8. For density map variance $\beta$, the MAE is relatively stable for different $\beta$, showing that our loss is relatively robust to this parameter. We also show L2 loss with large $\beta$ for different noise levels in Fig. 4. For large $\beta$, the performance of L2 is bad because the density map is overly smoothed.

### 4.2.4 Comparison of different preserved variances

We next evaluate the effectiveness of the correlation term and the low-rank approximation on the Mall [40] dataset. We use Mall because the crowd size in the image is small, so that we can fit the low-rank approximation with almost 100% of variance into the GPU memory. The quantitative results are shown in Fig. 6 and the visualization is shown in Fig. 8. The MAE is worse when less than 60% variance is selected, since some useful correlations are not considered. Interestingly, the MAE increases slightly when over 60% variance is selected. We surmise that this occurs because selecting more background pixels is equivalent to increasing the weight of background pixels, which puts more focus on background and hurts performance.

### 4.2.5 Comparison of each component

We conduct an experiment to investigate the effect of each component of our loss, and the results are in Tab. 2. "FullCov", which uses the full-covariance matrix, outperforms "DiagCov", which assumes independent Gaussians (diagonal covariance), confirming the importance of correlations between pixels. Adding the regularization term and background model further improves the results. For the methods without background model, we add a small constant to the variances to avoid infinite loss.

### 4.3 Comparison with state-of-the-arts

Finally, we compare VGG19 trained with our loss function with state-of-the-art algorithms on 5 datasets, and the results are shown in Tab. 3. First, our method uses the same backbone but achieves better performance than BL, which confirms the effectiveness of our loss function. Second, our method achieves the best MAE on most of the datasets, including the three large datasets NWPU-Crowd, JHU-CROWD++ and UCF-QNRF. On the smaller dataset ShanghaiTech_A and

B , DSSINet [16] is better than our method since it uses multi-scale images to extract features. Similarly, MBTTBF [43] achieves better performance on ShanghaiTech_A by fusing multi-level features together. However, those architectures do not generalize well to large-scale dataset such as JHU-CROWD++ and UCF-QNRF. Note that our VGG19 backbone does not do any special operation in fusing multi-scale features. We also use our loss function with the self-correlation strategy from [44], and achieve slightly improved performance of MAE 84.3 on UCF-QNRF. Finally, we compare our method with an uncertainty learning method [31], and achieve better performance as shown in Tab. 3.

## 5   Conclusion

In this paper, we propose a novel loss function based on the modeling of annotation noise, which is demonstrated to be robust to annotation noise. The loss function can be decomposes into a pixel-wise weighted MSE term, which focuses less on uncertain regions, and a correlation term, which models the correlation between pixels. We first defined the annotation noise as a r.v. with Gaussian distribution. Then, the probability density function of the density values is derived. We also propose to approximate the joint distribution of density values with a full covariance Gaussian and a low-rank approximation to reduce computational cost. The experimental results on 5 datasets confirm the effectiveness of the proposed loss function. Although in this paper we focus on crowd density maps, our derived loss function could also be used in other areas where response maps are constructed from noisy point-wise annotations, e.g., saliency maps, object detection maps, visual tracking response maps. Future work will explore these directions.

## Broader Impact

In this paper, we introduce a novel loss function for counting crowd numbers by explicitly considering annotation noise. It can be applied to any density map based network architecture and improve the counting accuracy generally. The research is also helpful for monitoring the crowd number in public and prevent the accidents caused by overcrowding. It could also be used in retail businesses to estimate the occupancy of a store or area, which helps with personal and resource management. Our method could also be applied to other objects, such as cell counting, plant/animal counting, etc, and other research areas that use point-wise annotations, e.g., eye gaze estimation.

Since the research is based on images captured by cameras, users may be concerned about the privacy problem. However, our method does not directly detect or track individuals, and thus this concern may be eased.

## Acknowledgments and Disclosure of Funding

There is no funding source to disclose for this paper.

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
