[Supplementary Material]

# Supplemental for Modeling Noisy Annotations for Crowd Counting

**Jia Wan**     **Antoni B. Chan**
Department of Computer Science
City University of Hong Kong
jiawan1998@gmail.com, abchan@cityu.edu.hk

## 1 Derivations

Here we provide the full derivations from the paper.

### 1.1 $\phi_i$ in probability distribution of $\Phi$

The individual term $\phi_i$ is

$$\phi_i = \frac{1}{2\pi\beta}\exp(-\frac{1}{2}||\mathbf{q}_i - \epsilon_i||^2_{\beta\mathbf{I}}), \tag{1}$$

which consists of a series of transformations of a multivariate Gaussian r.v.: squared L2 norm, negative exponential, and scaling.

**Mahalanobis distance**: To derive $\phi_i$, we first calculate the r.v. for the Mahalanobis distance $m_i$:

$$m_i = ||\mathbf{q}_i - \epsilon_i||^2_\beta = \frac{\alpha}{\beta}||\frac{1}{\sqrt{\alpha}}\mathbf{q}_i - \eta_0||^2 = \frac{\alpha}{\beta}||\eta_{\mathbf{r}_i}||^2, \tag{2}$$

where we define $\mathbf{r}_i = \frac{1}{\sqrt{\alpha}}\mathbf{q}_i$, and $\eta_\mu$ is a Gaussian with unit variance and mean $\mu$. The term $||\eta_{\mathbf{r}_i}||^2$ is a non-central $\chi^2$ r.v. with non-centrality parameter $\lambda_i = ||\mathbf{r}_i||^2$ and DoF $k = 2$ with pdf

$$\tilde{\chi}^2_{\lambda_i}(x) = \frac{1}{2}\exp(-\frac{x+\lambda_i}{2})I_0(\sqrt{\lambda_i x}), \tag{3}$$

where $I_0(x)$ is a modified beessel function of the 1st kind of order 0. When $\lambda_i = 0$, this simplifies to a standard $\chi^2$ distribution with 2 DoF,

$$\tilde{\chi}^2_0(x) = \chi^2(x) = \frac{1}{2}e^{-x/2}. \tag{4}$$

Finally, using the scaling property of r.v. $X$, $aX \sim \frac{1}{a}p(\frac{x}{a})$, we obtain the pdf of $m_i$,

$$m_i \sim p(m_i) = \delta\tilde{\chi}^2_{\lambda_i}(\delta m_i), \quad \delta = \frac{\beta}{\alpha}. \tag{5}$$

**Negative exponential**: Next, we derive the pdf of the negative exponential transformation of $m_i$. Consider the transformation of a random variable: $Y = g(X) = e^{-X/2}$. The inverse transformation is: $X = g^{-1}(Y) = -2\log Y$ with derivative $\frac{dg^{-1}}{dy} = \frac{-2}{y}$. Then the pdf of $Y$ is:

$$p(y) = p_x(g^{-1}(y))|\frac{dg^{-1}}{dy}| = \frac{2}{y}p_x(-2\log y). \tag{6}$$

Define the r.v. for the negative expo term $n_i = \exp(-\frac{1}{2}m_i)$ and applying the above transformation rule, then its pdf is:

$$n_i \sim p(n_i) = \frac{2\delta}{n_i}\tilde{\chi}^2_{\lambda_i}(-2\delta \log n_i). \tag{7}$$

**Scaling:** Finally, we have $\phi_i = hn_i$, where $h = (2\pi\beta)^{-1}$ which is the maximum value of one density kernel. Using the scaling property, we obtain the pdf of $\phi_i$:

$$\phi_i \sim p(\phi_i) = \frac{1}{h}n_i(\frac{1}{h}\phi_i) = \frac{2\delta}{h\phi_i}\tilde{\chi}^2_{\lambda_i}(-2\delta \log \frac{\phi_i}{h}). \tag{8}$$

Substituting for the non-central $\chi^2$ we obtain the explicit form of the pdf:

$$\phi_i \sim p(\phi_i) = \frac{2\delta}{h\phi_i}\left[\frac{1}{2}\exp(-\frac{-2\delta\log\frac{\phi_i}{h}+\lambda_i}{2})I_0(\sqrt{\lambda_i(-2\delta\log\frac{\phi_i}{h})})\right] \tag{9}$$

$$= \frac{\delta}{h}e^{-\lambda_i/2}\left(\frac{\phi_i}{h}\right)^{\delta-1}I_0(\sqrt{-2\delta\lambda_i\log\frac{\phi_i}{h}}) \tag{10}$$

When $\lambda_i = 0$ this simplifies to:

$$\phi_i \sim p(\phi_i) = \frac{\delta}{h}\left(\frac{\phi_i}{h}\right)^{\delta-1}. \tag{11}$$

The log-pdf (negative loss) is then

$$\log p(\phi_i) = \log\frac{\delta}{h} - \frac{\lambda_i}{2} + (\delta-1)\log\frac{\phi_i}{h} + \log I_0(\sqrt{-2\delta\lambda_i\log\frac{\phi_i}{h}}). \tag{12}$$

And when $\lambda_i = 0$,

$$\log p(\phi_i) = \log\frac{\delta}{h} + (\delta-1)\log(\frac{\phi_i}{h}). \tag{13}$$

## 1.2 $\mu$ and $\sigma^2$ for Gaussian approximation to $\Phi$

The mean of $\Phi$ is calculated as

$$\mathbb{E}[\Phi] = \mathbb{E}[\sum_i \phi_i] \tag{14}$$

$$= \mathbb{E}[\sum_i \mathcal{N}(\mathbf{q}_i|\boldsymbol{\epsilon}_i, \beta\mathbf{I})] \tag{15}$$

$$= \sum_i \int \mathcal{N}(\mathbf{q}_i|\boldsymbol{\epsilon}_i, \beta\mathbf{I})\mathcal{N}(\boldsymbol{\epsilon}_i|0, \alpha\mathbf{I})d\boldsymbol{\epsilon}_i \tag{16}$$

$$= \sum_i \underbrace{\mathcal{N}(\mathbf{q}_i|0, (\alpha+\beta)\mathbf{I})}_{\mu_i}, \tag{17}$$

where the annotation noise is $\epsilon_i \sim \mathcal{N}(0, \alpha\mathbf{I})$, and (17) follows from (53).

To calculate the variance of $\Phi$, we first compute the second moment:

$$\mathbb{E}[(\sum_i \phi_i)^2] = \sum_{ij}\mathbb{E}[\phi_i\phi_j] = \sum_i\mathbb{E}[\phi_i^2] + \sum_{i\neq j}\mathbb{E}[\phi_i\phi_j]. \tag{18}$$

There are two cases:

- When $i \neq j$, $\mathbb{E}[\phi_i\phi_j] = \mathbb{E}[\phi_i]\mathbb{E}[\phi_j] = \mu_i\mu_j$, since $\epsilon_i$ is independent of $\epsilon_j$.

- When $i = j$,

$$\mathbb{E}[\phi_i^2] = \mathbb{E}[\mathcal{N}(\mathbf{q}_i|\epsilon_i, \beta\mathbf{I})^2] \tag{19}$$

$$= \mathbb{E}[\mathcal{N}(\mathbf{q}_i|\mathbf{q}_i, 2\beta\mathbf{I})\mathcal{N}(\epsilon_i|\mathbf{q}_i, \frac{1}{2}\beta\mathbf{I})] \tag{20}$$

$$= \mathcal{N}(0|0, 2\beta\mathbf{I})\mathbb{E}[\mathcal{N}(\epsilon_i|\mathbf{q}_i, \frac{1}{2}\beta\mathbf{I})] \tag{21}$$

$$= \mathcal{N}(0|0, 2\beta\mathbf{I})\int \mathcal{N}(\epsilon_i|0, \alpha\mathbf{I})\mathcal{N}(\epsilon_i|\mathbf{q}_i, \frac{1}{2}\beta\mathbf{I})d\epsilon_i \tag{22}$$

$$= \mathcal{N}(0|0, 2\beta\mathbf{I})\mathcal{N}(\mathbf{q}_i|0, (\beta/2 + \alpha)\mathbf{I}) \tag{23}$$

$$= \frac{1}{4\pi\beta}\mathcal{N}(\mathbf{q}_i|0, (\beta/2 + \alpha)\mathbf{I}), \tag{24}$$

where (20) uses (54), and (23) follows from (53).

Next, we have

$$\mathbb{E}[\Phi]^2 = \sum_{ij} \mu_i\mu_j = \sum_i \mu_i^2 + \sum_{i\neq j} \mu_i\mu_j. \tag{25}$$

Finally, the variance is:

$$\text{var}(\Phi) = \sigma^2 = \mathbb{E}[\Phi^2] - \mathbb{E}[\Phi]^2 \tag{26}$$

$$= \sum_i \frac{1}{4\pi\beta}\mathcal{N}(\mathbf{q}_i|0, (\beta/2 + \alpha)\mathbf{I}) + \sum_{i\neq j}\mu_i\mu_j - \sum_i \mu_i^2 - \sum_{i\neq j}\mu_i\mu_j \tag{27}$$

$$= \sum_i \frac{1}{4\pi\beta}\mathcal{N}(\mathbf{q}_i|0, (\beta/2 + \alpha)\mathbf{I}) - \sum_i \mu_i^2. \tag{28}$$

## 1.3 Covariance for joint likelihood of $\Psi$

For the covariance, we first compute the second moment.

$$\mathbb{E}[\Phi^{(\eta)}\Phi^{(\rho)}] = \mathbb{E}[(\sum_i \phi_i^{(\eta)})(\sum_j \phi_j^{(\rho)})] = \sum_{ij}\mathbb{E}[\phi_i^{(\eta)}\phi_j^{(\rho)}] = \sum_i \mathbb{E}[\phi_i^{(\eta)}\phi_i^{(\rho)}] + \sum_{i\neq j}\mathbb{E}[\phi_i^{(\eta)}\phi_j^{(\rho)}]. \tag{29}$$

There are two cases:

- When $i \neq j$, $\epsilon_i$ and $\epsilon_j$ are independent, thus $\mathbb{E}[\phi_i^{(\eta)}\phi_j^{(\rho)}] = \mathbb{E}[\phi_i^{(\eta)}]\mathbb{E}[\phi_j^{(\rho)}] = \mu_i^{(\eta)}\mu_j^{(\rho)}$
- When $i = j$,

$$\omega_i^{(\eta,\rho)} = \mathbb{E}[\phi_i^{(\eta)}\phi_i^{(\rho)}] = \mathbb{E}[\mathcal{N}(\mathbf{q}_i^{(\eta)}|\epsilon_i, \beta\mathbf{I})\mathcal{N}(\mathbf{q}_i^{(\rho)}|\epsilon_i, \beta\mathbf{I})] \tag{30}$$

$$= \mathbb{E}[\mathcal{N}(\mathbf{q}_i^{(\eta)}|\mathbf{q}_i^{(\rho)}, 2\beta\mathbf{I})\mathcal{N}(\epsilon_i|\frac{1}{2}(\mathbf{q}_i^{(\eta)} + \mathbf{q}_i^{(\rho)}), \frac{\beta}{2}\mathbf{I})] \tag{31}$$

$$= \mathcal{N}(\mathbf{q}_i^{(\eta)}|\mathbf{q}_i^{(\rho)}, 2\beta\mathbf{I})\mathbb{E}[\mathcal{N}(\epsilon_i|\frac{1}{2}(\mathbf{q}_i^{(\eta)} + \mathbf{q}_i^{(\rho)}), \frac{\beta}{2}\mathbf{I})] \tag{32}$$

$$= \mathcal{N}(\mathbf{q}_i^{(\eta)}|\mathbf{q}_i^{(\rho)}, 2\beta\mathbf{I})\mathcal{N}(\frac{1}{2}(\mathbf{q}_i^{(\eta)} + \mathbf{q}_i^{(\rho)})|0, (\beta/2 + \alpha)\mathbf{I}) \tag{33}$$

$$= \mathcal{N}(\mathbf{x}^{(\eta)}|\mathbf{x}^{(\rho)}, 2\beta\mathbf{I})\mathcal{N}(\frac{1}{2}(\mathbf{q}_i^{(\eta)} + \mathbf{q}_i^{(\rho)})|0, (\beta/2 + \alpha)\mathbf{I}), \tag{34}$$

where (31) follows from (57), (33) uses (53), and (34) uses

$$\mathbf{q}_i^{(\eta)} - \mathbf{q}_i^{(\rho)} = (\mathbf{x}^{(\eta)} - \tilde{\mathbf{D}}_i) - (\mathbf{x}^{(\rho)} - \tilde{\mathbf{D}}_i) = \mathbf{x}^{(\eta)} - \mathbf{x}^{(\rho)}. \tag{35}$$

Thus, the second moment is:

$$\mathbb{E}[\Phi^{(\eta)}\Phi^{(\rho)}] = \sum_i \omega_i^{(\eta,\rho)} + \sum_{i\neq j}\mu_i^{(\eta)}\mu_j^{(\rho)} \tag{36}$$

Finally, the covariance is:

$$\text{cov}(\Phi^{(\eta)}, \Phi^{(\rho)}) = \mathbb{E}[\Phi^{(\eta)}\Phi^{(\rho)}] - \mathbb{E}[\Phi^{(\eta)}]\mathbb{E}[\Phi^{(\rho)}] \tag{37}$$

$$= \sum_i \omega_i^{(\eta,\rho)} + \sum_{i \neq j} \mu_i^{(\eta)}\mu_j^{(\rho)} - \left(\sum_i \mu_i^{(\eta)}\right)\left(\sum_j \mu_j^{(\rho)}\right) \tag{38}$$

$$= \sum_i \omega_i^{(\eta,\rho)} + \sum_{i \neq j} \mu_i^{(\eta)}\mu_j^{(\rho)} - \sum_{ij} \mu_i^{(\eta)}\mu_j^{(\rho)} \tag{39}$$

$$= \sum_i \omega_i^{(\eta,\rho)} - \sum_i \mu_i^{(\eta)}\mu_i^{(\rho)}. \tag{40}$$

## 1.4 Low-rank approximation to covariance matrix

Assume there are $P$ spatial locations, and $\boldsymbol{\mu}, \boldsymbol{\Sigma}$ is the mean and full covariance matrix of the $\boldsymbol{\Psi}$s. We first make the following definitions:

- Let $\mathbf{v}$ be a $P$-dim vector of variances $\text{var}(\Phi^{(\eta)})$. This is the same as the variances when assuming the $\Phi$s are independent, and $\mathbf{V} = \text{diag}(\mathbf{v})$.

- Let $\mathcal{M} = \{m_1, \cdots, m_M\}$ be the set of indices of spatial locations $\mathbf{x}^{(m_i)}$ that we want to use for the approximation.

- Let $\mathbf{m}$ be a $P$-dim vector mask corresponding to $\mathcal{M}$.

- Let $\mathbf{v}_{\mathcal{M}}$ be the vector of variances for indices in $\mathcal{M}$.

- Let $\mathbf{M}$ be the "permutation" matrix that selects items in $\mathbf{v}$ to construct $\mathbf{v}_{\mathcal{M}}$. I.e., the i-th column $[\mathbf{M}]_i = \mathbf{e}_{m_i}$, where $\mathbf{e}_i$ is the i-th canonical unit vector. Note that $\mathbf{M}^T\mathbf{M} = \mathbf{I}$, and $\mathbf{M}\mathbf{M}^T = (\mathbf{m}\mathbf{m}^T) \circ \mathbf{I}$ is the masked identity matrix.

- The vectors/matrices corresponding to the selected locations are: $\mathbf{v}_{\mathcal{M}} = \mathbf{M}^T\mathbf{v}$, and $\mathbf{V}_{\mathcal{M}} = \text{diag}(\mathbf{v}_{\mathcal{M}}) = \mathbf{M}^T\mathbf{V}\mathbf{M}$.

- The masked variances are $\mathbf{m} \circ \mathbf{v} = \mathbf{M}\mathbf{v}_{\mathcal{M}}$, where $\circ$ is element-wise product

- The masked diagonal matrix is $(\mathbf{m}\mathbf{m}^T) \circ \mathbf{V} = \mathbf{M}\mathbf{V}_{\mathcal{M}}\mathbf{M}^T$.

- Define the matrix $\mathbf{A}$ of off-diagonal covariances as:

$$[\mathbf{A}]_{ij} = \begin{cases} 0, & i = j \\ \text{cov}(\Phi^{(\eta_i)}, \Phi^{(\eta_j)}), & i \neq j \end{cases}$$

- Define the matrix $\mathbf{A}_{\mathcal{M}}$ that selects rows/columns of $\mathbf{A}$ according to $\mathcal{M}$. Thus, $\mathbf{A}_{\mathcal{M}} = \mathbf{M}^T\mathbf{A}\mathbf{M}$ and the masked $\mathbf{A}$ matrix is $(\mathbf{m}\mathbf{m}^T) \circ \mathbf{A} = \mathbf{M}\mathbf{A}_{\mathcal{M}}\mathbf{M}^T$.

Using the matrices corresponding to the diagonal and off-diagonal elements, the covariance matrix is $\boldsymbol{\Sigma} = \mathbf{V} + \mathbf{A}$. We obtain an approximation by only using the off-diagonal elements corresponding to $\mathcal{M}$, giving

$$\hat{\boldsymbol{\Sigma}} = \mathbf{V} + \mathbf{M}\mathbf{A}_{\mathcal{M}}\mathbf{M}^T.$$

Using the matrix inversion lemma, we obtain the approximate precision matrix

$$\hat{\boldsymbol{\Sigma}}^{-1} = \mathbf{V}^{-1} - \mathbf{V}^{-1}\mathbf{M}(\mathbf{A}_{\mathcal{M}}^{-1} + \mathbf{M}^T\mathbf{V}^{-1}\mathbf{M})^{-1}\mathbf{M}^T\mathbf{V}^{-1} \tag{41}$$

$$= \mathbf{V}^{-1} - \mathbf{V}^{-1}\mathbf{M}(\mathbf{A}_{\mathcal{M}}^{-1} + \mathbf{V}_{\mathcal{M}}^{-1})^{-1}\mathbf{M}^T\mathbf{V}^{-1} \tag{42}$$

$$= \mathbf{V}^{-1} - \mathbf{V}^{-1}\mathbf{M}\mathbf{V}_{\mathcal{M}}(\mathbf{V}_{\mathcal{M}}\mathbf{A}_{\mathcal{M}}^{-1}\mathbf{V}_{\mathcal{M}} + \mathbf{V}_{\mathcal{M}})^{-1}\mathbf{V}_{\mathcal{M}}\mathbf{M}^T\mathbf{V}^{-1} \tag{43}$$

$$= \mathbf{V}^{-1} - \mathbf{M}\underbrace{(\mathbf{V}_{\mathcal{M}}\mathbf{A}_{\mathcal{M}}^{-1}\mathbf{V}_{\mathcal{M}} + \mathbf{V}_{\mathcal{M}})^{-1}}_{\mathbf{B}_{\mathcal{M}}}\mathbf{M}^T = \mathbf{V}^{-1} - \mathbf{M}\mathbf{B}_{\mathcal{M}}\mathbf{M}^T, \tag{44}$$

where the last line follows from

$$\mathbf{V}^{-1}\mathbf{M}\mathbf{V}_{\mathcal{M}} = \mathbf{V}^{-1}\mathbf{M}\mathbf{M}^T\mathbf{V}\mathbf{M} = \mathbf{M} \tag{45}$$

The final approximation of the inverse covariance is $\hat{\boldsymbol{\Sigma}}^{-1} = \mathbf{V}^{-1} - \mathbf{M}\mathbf{B}_{\mathcal{M}}\mathbf{M}^T$, where $\mathbf{B}_{\mathcal{M}} = (\mathbf{V}_{\mathcal{M}}\mathbf{A}_{\mathcal{M}}^{-1}\mathbf{V}_{\mathcal{M}} + \mathbf{V}_{\mathcal{M}})^{-1}$.

Let $\boldsymbol{\Psi}$ be the vectorized predicted density map, and the error vector is $\bar{\boldsymbol{\Psi}} = \boldsymbol{\Psi} - \boldsymbol{\mu}$, where $\boldsymbol{\mu}$ is the mean vector. The Mahalonobis distance term of the Gaussian is:

$$\bar{\boldsymbol{\Psi}}^T \boldsymbol{\Sigma}^{-1} \bar{\boldsymbol{\Psi}} \approx \bar{\boldsymbol{\Psi}}^T \hat{\Sigma}^{-1} \bar{\boldsymbol{\Psi}} \tag{46}$$

$$= \bar{\boldsymbol{\Psi}}^T \mathbf{V}^{-1} \bar{\boldsymbol{\Psi}} - \bar{\boldsymbol{\Psi}}^T \mathbf{M} \mathbf{B}_{\mathcal{M}} \mathbf{M}^T \bar{\boldsymbol{\Psi}}. \tag{47}$$

The first term is the likelihood term when assuming independent Gaussians:

$$\bar{\boldsymbol{\Psi}}^T \mathbf{V}^{-1} \bar{\boldsymbol{\Psi}} = \sum_i \frac{1}{v_i} \bar{\Psi}_i^2. \tag{48}$$

The second term selects the portion of $\bar{\boldsymbol{\Psi}}$ belonging to $\mathcal{M}$ and computes a quadratic,

$$\bar{\boldsymbol{\Psi}}^T \mathbf{M} \mathbf{B}_{\mathcal{M}} \mathbf{M}^T \bar{\boldsymbol{\Psi}} = \bar{\boldsymbol{\Psi}}_{\mathcal{M}}^T \mathbf{B}_{\mathcal{M}} \bar{\boldsymbol{\Psi}}_{\mathcal{M}}, \tag{49}$$

where $\bar{\boldsymbol{\Psi}}_{\mathcal{M}}$ selects indices $\mathcal{M}$ from $\bar{\boldsymbol{\Psi}}$.

## 1.5 Useful properties

Here we list some useful properties regarding products of Gaussian distributions and their integrals:

1. Property 1:

$$\mathcal{N}(\mathbf{x}|\mathbf{a}, \mathbf{A})\mathcal{N}(\mathbf{x}|\mathbf{b}, \mathbf{B}) = \mathcal{N}(\mathbf{a}|\mathbf{b}, \mathbf{A} + \mathbf{B})\mathcal{N}(\mathbf{x}|\mathbf{c}, \mathbf{C}), \tag{50}$$

$$\mathbf{c} = \mathbf{C}(\mathbf{A}^{-1}\mathbf{a} + \mathbf{B}^{-1}\mathbf{b}), \tag{51}$$

$$\mathbf{C} = (\mathbf{A}^{-1} + \mathbf{B}^{-1})^{-1}. \tag{52}$$

2. Property 2:

$$\int \mathcal{N}(\mathbf{x}|\mathbf{a}, \mathbf{A})\mathcal{N}(\mathbf{x}|\mathbf{b}, \mathbf{B})d\mathbf{x} = \mathcal{N}(\mathbf{a}|\mathbf{b}, \mathbf{A} + \mathbf{B}) \tag{53}$$

3. Property 3:

$$\mathcal{N}(\mathbf{x}|\mathbf{a}, \mathbf{A})^2 = \mathcal{N}(\mathbf{a}|\mathbf{a}, 2\mathbf{A})\mathcal{N}(\mathbf{x}|\mathbf{a}, \frac{1}{2}\mathbf{A}), \tag{54}$$

$$\mathbf{C} = (2\mathbf{A}^{-1})^{-1} = \frac{1}{2}\mathbf{A}, \tag{55}$$

$$\mathbf{c} = \frac{1}{2}\mathbf{A}(2\mathbf{A}^{-1}\mathbf{a}) = \mathbf{a}. \tag{56}$$

4. Property 4:

$$\mathcal{N}(\mathbf{x}|\mathbf{a}_1, \mathbf{A})\mathcal{N}(\mathbf{x}|\mathbf{a}_2, \mathbf{A}) = \mathcal{N}(\mathbf{a}_1|\mathbf{a}_2, 2\mathbf{A})\mathcal{N}(\mathbf{x}|\frac{1}{2}(\mathbf{a}_1 + \mathbf{a}_2), \frac{1}{2}\mathbf{A}), \tag{57}$$

$$\mathbf{C} = (2\mathbf{A}^{-1})^{-1} = \frac{1}{2}\mathbf{A}, \tag{58}$$

$$\mathbf{c} = \frac{1}{2}\mathbf{A}(\mathbf{A}^{-1}\mathbf{a}_1 + \mathbf{A}^{-1}\mathbf{a}_2) = \frac{1}{2}(\mathbf{a}_1 + \mathbf{a}_2). \tag{59}$$

# 2 Visualization of density maps

We compare density maps predicted from models trained with different loss functions and noise levels in Fig. 1.

Figure 1: Visualization of density maps predicted from models trained with different loss functions and noise levels. (yellow arrow) the annotations are not at the center of heads and our method can move those dots toward heads' center, while BL cannot move them and L2 is less confident. (white dash ellipse) other methods have more false positive than ours. (black solid ellipses) when there is large annotation noise, the head region of our prediction is larger than the surrounding background, while others are the opposite. (red rectangle) since the annotation is very noisy, other method are confused about the foreground and background regions, while our prediction is roughly correct and smoother than others.