[Reviews · NeurIPS 2020]

Review 1

Summary and Contributions: In this work, the authors propose to model the annotation noise in crowd counting with Gaussian distribution. The noise arising from human annotation in Gaussian kernel generated density map is not i.i.d thus the correlation between pixels should be considered. By minimizing the negative log-likelihood of the approximated joint distribution of the density map values with a full covariance multivariate Gaussian density, the model has the flexibility to handle the annotation noise and achieve competitive performance in crowd counting.

Strengths: 1. This work first considers the noise in point-wise annotation in crowd counting which is a practical problem worth noting. 2. The proposed modeling method and the proposed loss function are novel. 3. The experiments demonstrate the effectiveness of the proposed methods achieve competitive performance for different networks in different dataset.

Weaknesses: Missing references. uncertianty learning has been well studied and related work should be discussed. Further, this paper should be compared with uncetainty learning methods. The reason that the distribution of \Phi which is the summation of individual {\chi}^2 distribution is approximated with Gaussian distribution is unclear. The comparison in Tab. 2 should include “L2 + Reg (L_i)” which is a reasonable comparison. The effect of \alpha and \beta should be jointly considered. Large \beta is believed to be helpful to handle the annotation noise, which reduces the peak value. It should be studied the effect of \alpha when \beta is large. Is it still effective? The robustness of different \beta in L2 loss should also be evaluated like Fig.4. It sould be better to further verify the proposed method on other tasks for example key-point estimation to demonstrate the generalizability. Post Rebuttal: The uncertainty learning based baseline is not convincing. Further, it would be much better to show the generalization of the proposed method on other tasks.

Correctness: It is unclear why the authors approximate the distribution \Phi which is the summation of individual {\chi}^2 distribution with Gaussian distribution.

Clarity: Yes

Relation to Prior Work: Yes.

Reproducibility: Yes

Additional Feedback:


Review 2

Summary and Contributions: This paper aims at addressing the issue of annotation noise in crowd counting, which is ignored by most of traditional density map based crowd counting methods. A new loss function based on the modeling of annotation noise is proposed, which can be decomposed into a pixel wise weighted MSE term and a correlation term. Extensive experimental results have demonstrated the effectiveness of the designed loss function towards the robustness of annotation noise for crowd counting.

Strengths: The idea of improving crowd counting from the perspective of modeling the annotation noise is novel. The approximation of the loss function which reduces its computational cost seems ingenious. Experimental results as well as detailed ablation studies have demonstrated the effectiveness of the proposed loss function.

Weaknesses: (1) How the correlation between pixels is modeled in the proposed loss function is not very clear. (2) The authors mentioned that 3 backbones including VGG19 [4], CSRNet [1], and MCNN [11] were tested in Section 4.1, but only VGG19 results were reported. Please correct me if I have missed the details. (3) Since the proposed loss function is universal, it needs to be integrated into the existing SOTA crowd counting algorithms to better verify its effectiveness. (4) The related work part simply lists the existing methods, and does not comment on these methods or explain the relationship between these methods and the proposed one. (5) The paper is poorly written, and there are a lot of grammatical errors and typos that make the paper hard to follow. (6) I don’t quite understand Line 119~Line 123 “However, such an assumption is tenuous, as the observation noise is induced by the noise in the observed annotation locations D that is passed through the non-linear function in (1).” (7) I don't quite follow Line 26 "However, if an annotation moves, the changes in the density map in nearby pixels are correlated".

Correctness: Seems correct.

Clarity: The paper is poorly written, and there are a lot of grammatical errors and typos that make the paper hard to follow. To name a few as follows: (1) Line 55 "direct" should be “ directly” (2) Line 70 and 71 “propose” should be “proposes” (3) Line 256 “ showing that are loss is relatively robust to this parameter” (4) Line 262 "since the some correlations are not considered”

Relation to Prior Work: Yes

Reproducibility: Yes

Additional Feedback: I have read the authors' feedback and other reviewers' comments. I tend towards maintaining my original rating.


Review 3

Summary and Contributions: The paper focusses on the problem of crowd counting. Specifically, the authors model annotation noise using random variable with Gaussian Distribution and they model the pdf of the crowd density value at each location in the density map. The resulting loss function leads to robust results in the presence of simulated noise. Also, they are able to achieve better results compared to recent methods on several datasets.

Strengths: 1. The problem of modeling annotation noise has not been explored earlier in crowd counting. This is a promising direction and an important problem. 2. Empirical results show that the proposed method is more robust in terms of overall error in the presence of simulated noise. Also, the proposed loss function achieves better results compared to recent SOTA. 3. The authors have conducted extensive experiments on multiple datasets.

Weaknesses: 1. A major concern that I have is, the authors consider only shifts in annotations as the noise. However, real-world annotations include other types of noises like missing annotations or duplicate annotations. The authors do not consider this in their discussion. From the outset, it seems that their current method cannot accommodate these additional noises. From this perspective, I would say that the paper is incomplete in modeling different types of annotation noises. 2. It is not clear why the authors approximate pdf phi and Psi with Gaussian distribution. 3. It is not clear why eta_ri term is non-central chi-squared distribution. 4. As far as I understand, small shifts in annotations will not affect performance much, since neural networks can be robust if receptive size of the network is large enough. Can the authors discuss this more in detail. 5. The proposed method seems to be too specific to the counting problem. Can this method be extended to other problems in vision like object detection.

Correctness: May be

Clarity: Somewhat. More discussions and clarifications are required. See weaknesses section.

Relation to Prior Work: Yes

Reproducibility: Yes

Additional Feedback: The authors have partially addressed my concerns. However, I do not agree that duplicate annotations are rare - being involved in data collection efforts - this is not uncommon, especially for pointwise annotations. I upgrade my rating slighlty.


Review 4

Summary and Contributions: This paper proposes a new method for capturing noisy annnotation in crowd counting. Instead of treating the point annotation as the "true" location of a person head, this paper assumes it is a noisy version of some underlying true location with Gaussian noisy. Based on this formulation, this paper proposes a loss function for crowd counting.

Strengths: The idea of modeling annotation noise in the context of crowd counting is relatively new. Based on the experiments, the proposed loss (which captures the annotation noise) seems to work well compared with existing loss function used in crowd counting.

Weaknesses: The novelty of the paper is a bit limited. The idea of modeling annoation noise has been explored in [4] (although in sightly different form). The general idea of learning from noisy labels has also been explored in many previous works. The main contribution of the paper is use this idea in the specific context of crowd counting.

Correctness: Seems to be correct, although I did not fully verify all the math derivations

Clarity: Yes

Relation to Prior Work: Previous works on learning from noisy annotation should be expanded. The difference with these works should be more elaborated.

Reproducibility: Yes

Additional Feedback: None

[Author Response · NeurIPS 2020]

We thank the reviewers for the insightful comments, and we will update the paper accordingly.

**R1Q1:** compare with uncertainty learning methods. **A:** Using the loss from ["What Uncertainties Do We Need in Bayesian Deep Learning for Computer Vision?", NeurIPS 2017] on UCF-QNRF we obtain 103.2/168.2 (MAE/MSE), which is worse than our proposed loss. We will add more references about uncertainty learning.

**R1Q2, R3Q2:** Why $\Phi$ and $\Psi$ are approximated with Gaussian? **A:** Using Gaussians for approximate inference is common, since they are tractable and can be estimated from 1st and 2nd moments. Extensions of the central limit theorem prove that sums of independent non-identical r.v.s converge to Gaussian. Indeed, in Fig 2c, the distribution is tending to Gaussian with just 3 annotations, and we observe this tendency becomes stronger with more annotations. We have also tried Gamma distributions for the approximation, but the results are worse (MAE 89.7 UCF-QNRF).

**R1Q3:** Tab. 2 should include "L2+Reg ($L_i$)". **A:** The result of L2+Reg is 94.5/160.0, which is worse than our loss.

**R1Q4:** The effect of $\alpha$ when $\beta$ is large? Robustness of L2 to different $\beta$. **A:** We evaluate the effect of $\alpha$ when $\beta = 16$ in Fig. R1, and the proposed loss is effective for a larger $\beta$. We also show L2 loss with large $\beta$ for different noise levels in Fig. R2. For large $\beta$, the performance is bad because the density map is over-smoothed.

**R1Q5, R3Q5, R4:** Try on other tasks to show generalizability? **A:** We propose a general framework to model noise in point-wise annotations that are converted to response maps. In future work, we will investigate other tasks that use response maps, such as human joint detection and visual object tracking. However, we think the derivation of the proposed framework and approximation method, as

Fig. R1: large $\beta$.    Fig. R2: Annot. noise.

well as extensive experiments on crowd counting, are strong enough for a standalone paper that can inspire others.

**R2Q1:** How correlation between pixels is modeled? **A:** Each dimension of the m.v. Gaussian corresponds to one pixel in the density map, and thus the covariance matrix models the correlations between pixels (see L157-169, Figs. 2 & 3). Equivalently, the loss is Mahalanobis distance (Eq. 10).

**R2Q2:** Other backbones CSRNet and MCNN? **A:** The performance of different backbones is reported in Table 1 (Sec. 4.2.1). Since VGG19 works better in the ablation, we only show VGG19 in the remaining experiments.

**R2Q3:** Integrate proposed loss with existing SOTA crowd counting? **A:** VGG19 is the strongest backbone before 2020, and we evaluate other backbones in Table 1. We also use our loss function with self-correlation strategy ["Adaptive Dilated Network with Self-Correction Supervision for Counting", CVPR 2020] (*published after the NeurIPS submission deadline*), and achieve better performance on UCF-QNRF (84.3/142.9 vs. 85.8/150.6).

**R2Q4:** Missing discussion between related and proposed works. **A:** See L82-84 and L93-98 for these discussions.

**R2Q5:** grammatical errors and typos. **A:** We will revise the paper.

**R2Q6:** Explain L119-123. **A:** i.i.d Gaussian noise (L2 norm) assumes independent noise between pixels. However, a noisy annotation actually induces correlated noise between pixels in the density map, see R2Q7 for details.

**R2Q7:** Explain L26. **A:** If noise is added to the annotation, then the density map values in nearby pixels change in a common way (i.e., correlated). For example, in Fig 2a, if the right-most green dot moves towards $x^{(0)}$, then the density values at $x^{(0)}$ and $x^{(1)}$ will both increase. Also, if it moves away, then density values at $x^{(0)}$,$x^{(1)}$ will both decrease.

**R3Q1:** Does not handle missing or duplicate annotations. **A:** Missing annotations are handled by the background model (L197-203), which adds a "virtual dot" close to each pixel (equivalent to a hypothetical missing annotation). Duplicate annotations are rare, and will be corrected by the annotator in the review phase. Modeling displacement noise already yields substantial benefits, and we will investigate other kinds of annotation noise in future work.

**R3Q3:** Why $\eta_{\mathbf{r}_i}$ is non-central $\chi^2$? **A:** $\eta_{\mathbf{r}_i}$ is a m.v. Gaussian with mean $\mathbf{r}_i$ and identity covariance. $\|\eta_{\mathbf{r}_i}\|^2$ is the sum squares of Gaussian r.v.s with non-zero mean and unit variance, and thus a non-central $\chi^2$ distribution (by definition).

**R3Q4:** Small shifts in annotations will not affect performance much, since NN can be robust if receptive size of the network is large enough. **A:** Shifts in annotations cause changes in the GT *target output*, and it is more appropriate to modify the loss function in order to make the network less sensitive to these output changes (see Figs. 4 & 7). On the other side, using large receptive field and pooling layers will make the network invariant to local shifts in the *input image*, but this does not make it better at handling ambiguous outputs.

**R4Q1:** Novelty is limited; modeling annotation noise explored in [4]. **A:** Our approach is quite different from [4]. [4] uses the annotations to compute weights on the density map pixels, whereas our approach is a generative model mapping annotation noise to density map noise. [4] handles each annotation separately, while our work models correlations induced by multiple annotations (L148-149).

[Meta-Review · NeurIPS 2020]

This paper was reviewed by four knowledgeable experts, none of whom have particularly strong opinions in either direction about the work. The main concerns of R1 regard the uncertainty learning baseline (which, in my opinion, the rebuttal addresses adequately, especially given the limited time), and extension to other tasks. I agree with the authors that extension of this approach to new tasks is more appropriate in a separate work, given the novelty of the approach and the application to crowd counting. The concerns of R2 mainly regard clarity of writing, which after close study I do not find to be so distracting -- although there is room for improvement. I find this paper to be a novel approach addressing a problem that is usually completely ignored in the crowd counting literature, and thus I think it is a refreshing break from the current trend in the area -- and a fairly detailed study of the problem of modeling annotation noise in crowd counting data (even if not *all* types of noise). As such, I think NeurIPS is a perfect venue for just such a paper and my final decision is to accept.